# Diagnostic Value of Simple Ultrasound Features and Inflammatory Markers in Postmenopausal Ovarian Cysts

**DOI:** 10.3390/diagnostics15172220

**Published:** 2025-09-01

**Authors:** Balazs Erdodi, Gergo Jozsef Szollosi, David Ratonyi, Laszlo Varadi, Zoard Tibor Krasznai, Attila Jakab

**Affiliations:** 1Department of Obstetrics and Gynecology, Faculty of Medicine, University of Debrecen, 4032 Debrecen, Hungary; ratonyi.david@med.unideb.hu (D.R.); krasznai.zoard@med.unideb.hu (Z.T.K.); ja@med.unideb.hu (A.J.); 2Doctoral School of Clinical Sciences, University of Debrecen, 4032 Debrecen, Hungary; 3Coordination Center for Research in Social Sciences, Faculty of Economics and Business, University of Debrecen, 4032 Debrecen, Hungary; szollosi.gergo@econ.unideb.hu; 4Szent Margit Hospital, 1032 Budapest, Hungary; dr.varadi.laszlo@sztmargit.hu

**Keywords:** inflammatory marker, ovary, ovarian cancer, ultrasonography, complex cyst, simple cyst, postmenopause, diagnostics, basic ultrasound scan, NLR, PLR

## Abstract

**Background:** The management of ovarian cysts in postmenopausal women is still a diagnostic dilemma. Although ultrasound is the diagnostic cornerstone of the initial assessment, it is limited by its interpretation in cases without clear morphological features of malignancy. **Objectives:** The aim of this study was to assess whether the addition of grayscale ultrasound features with inflammatory markers including the neutrophil-to-lymphocyte ratio (NLR) and platelet-to-lymphocyte ratio (PLR) can improve diagnostic accuracy in the identification of malignant ovarian lesions as compared to benign cysts in postmenopausal women. **Methods:** A total of 103 surgically removed adnexal masses were examined retrospectively. Ultrasound morphology was categorized to either simple or complex while NLR and PLR were calculated from preoperative full blood counts. The reference standard was histopathology. **Results:** Of the 103 cysts taken out, 74 cysts (71.8%) were benign while 29 cysts (28.2%) were malignant. Complex morphology was shown by all malignant lesions. NLR values in malignancy vs. benignancy showed a mean NLR of 4.96 ± 2.3 in the malignant cases, while it was 2.56 ± 1.2 in the benign cases (*p* < 0.001). In a similar fashion, the PLR was 198.4 ± 45.1 in malignant compared to 134.2 ± 32.7 in benign cases (*p* < 0.001). In the group of complex cysts (*n* = 52), NLR and PLR were compared to differentiate between malignant and benign lesions. In logistic regression, complex morphology was an independent predictor of malignancy, while NLR showed a positive, non-significant trend; PLR was not independently associated. **Conclusions:** Use of NLR and PLR in combination with grayscale ultrasonographic morphology improves the diagnostic characterization of postmenopausal women with adnexal masses. This easy, cost-effective method might aid in better triage and surgery planning.

## 1. Introduction

The prevalence and clinical relevance of ovarian cysts in postmenopausal women present a diagnostic dilemma for gynecologists worldwide. As imaging technologies have advanced, the detection of asymptomatic adnexal masses has increased. The diagnosis of these pelvic masses was traditionally based on palpation findings, but as a consequence of the progress and technical improvements, transvaginal ultrasound (TVUS) became the gold standard method in the diagnosis of adnexal lesions. Since its introduction in the 1980s, it has become more detailed and sophisticated as image quality has increased in an excessive manner and this progress is still ongoing. Hundreds of investigations have been performed on supporting procedures, laboratory parameters, or even demographical data that could increase the diagnostic accuracy of TVUS.

Accurately distinguishing between benign and malignant pelvic masses is critical for appropriate clinical management. While both functional and malignant ovarian cysts may present with overlapping symptoms, establishing the underlying nature of these lesions is of paramount importance—second only to their early detection. Unfortunately, more than 60% of ovarian cancers are still diagnosed at advanced stages (FIGO stage III or IV), which substantially compromises survival outcomes [1,2,3]. Although a standardized screening protocol for ovarian cancer has yet to be established, emerging approaches increasingly support individualized risk assessment models and validated imaging techniques, as emphasized by Nebgen et al. [4]. Among serum biomarkers, cancer antigen 125 (CA-125) has historically been used to enhance detection rates, especially when interpreted alongside transvaginal ultrasonography [5,6,7]. More recently, human epididymis protein 4 (HE4) has shown potential as a complementary marker. However, despite its promise, HE4 has not significantly altered diagnostic strategies in expert settings, although it may provide added value for less experienced practitioners [8].

Since the advent of high-resolution transvaginal ultrasound, several sonographic scoring systems have been introduced to improve the diagnostic accuracy of grayscale imaging [9,10,11,12,13]. Notably, recent comparative studies such as that by Hiett et al. have assessed the performance of IOTA Simple Rules, the ADNEX model, and O-RADS in contemporary North American populations [14]. Specific morphological features—such as papillary projections, irregular or thickened walls, septations, and multilocular structures—are associated with a higher likelihood of malignancy. These features define the so-called complex cysts [15,16,17]. In a recent paper by our team, it was confirmed that complex cyst morphology confers a significantly increased risk of malignancy, particularly in postmenopausal women [18].

The International Ovarian Tumour Analysis (IOTA) group has played a pivotal role in developing evidence-based scoring systems through large-scale, multicenter studies [13,19,20]. Over the past two decades, IOTA has introduced several mathematical models and diagnostic frameworks that assist clinicians in differentiating benign from malignant adnexal lesions—regardless of the examiner’s ultrasound experience [21].

The tumor microenvironment plays an important role in tumor formation, spread, and malignant characteristics of neoplasms [22]. In this extensive cellular network, inflammatory cells are present, which play a leading role in the formation of a susceptible environment to the tumor cells. This is the reason why measurable serum parameters such as leukocyte count, platelet count, and C-reactive protein (CRP) are increased while albumin levels are decreased in patients carrying malignancy. Instead of using single blood parameters, composite markers such as the platelet-to-lymphocyte ratio (PLR) and neutrophil-to-lymphocyte (NLR) are rather preferred in clinical settings due to their increased stability and sensitivity [23]. Yun et al. reported that malignant ovarian tumors exhibit significantly higher NLR and PLR values than benign or borderline lesions. They established diagnostic cut-off values of 2.36 for NLR and 150.02 for PLR, emphasizing the potential of these inflammatory markers in stratifying ovarian masses prior to invasive procedures [24]. It has been reported that high preoperative NLR predicts poor overall survival [25], a finding confirmed in a recent meta-analysis by Bizon et al., which emphasized PLR as a highly sensitive diagnostic and prognostic marker in ovarian cancer [26], and also having predictive value in chemotherapeutic response [27]. Despite these favorable parameters, a uniform cut-off value has not yet been determined. According to a recent meta-analysis by Zhang et al., a wide range of cut-offs is reported but the majority are between 2 and 4. [28]. Just like NLR, higher PLR values also reflect poor survival in case of ovarian malignancy. Most papers on inflammatory markers focus on their effect on survival and therapeutic prediction roles, but only a few mention diagnostic support. However, diagnostic utility is increasingly supported; recently, Zhang et al. provided a synthesis of NLR/PLR cut-offs in diagnostic stratification, underlining their practical application [28]. Regarding the diagnosis of ovarian malignancy, Yildirim et al. reported an effective predictive role for PLR [29,30]; however, standardized cut-offs have not been reported for this marker by their team. A recent study by Huang et al. highlighted that PLR can be helpful in distinguishing borderline epithelial tumors from benign epithelial lesions, while its combination with NLR and CA125 can increase the detection of malignant epithelial tumors [31]. In addition to these findings, Song et al. conducted a comprehensive comparison of various systemic inflammatory indicators, including NLR and PLR, in the early diagnosis of ovarian cancer [32]. Their study highlighted that these markers, when used along with conventional biomarkers, could enhance diagnostic accuracy. This multi-marker approach reinforces the idea that a combination of inflammatory indices with established markers may better capture the complex biological processes driving ovarian tumorigenesis. Moreover, Ye et al. undertook an analysis that compared the diagnostic performance of NLR, PLR, and other inflammatory and tumor markers [33]. Their findings indicate that these ratios could distinguish ovarian cancer patients from those with benign ovarian conditions, thereby supporting them as potential non-invasive biomarkers. Complementing these studies, Li et al. evaluated preoperative inflammatory markers and found that NLR and PLR could effectively distinguish epithelial ovarian cancer from benign ovarian masses [34]. This study underscores the clinical utility of these markers in the diagnostic workflow and highlights the necessity for integrating inflammatory markers with other diagnostic assessments to improve patient stratification. Moreover, in resource-limited settings where expert sonographers or advanced imaging are not always available, the integration of such systemic markers may increase the reliability of triage decisions [35].

Additionally, known risk factors—such as genetic predisposition, parity, and prior gynecological surgery—have been linked to the development of ovarian cysts [36,37].

We evaluated whether systemic inflammatory indices—the neutrophil-to-lymphocyte ratio (NLR) and platelet-to-lymphocyte ratio (PLR)—add diagnostic value to simple grayscale ultrasound morphology (simple vs. complex) for distinguishing benign from malignant ovarian cysts in postmenopausal women. Building on our previous observation that complex cyst morphology is a strong predictor of malignancy in this population [18], we prespecified subgroup analyses by morphology and size to enhance clinical interpretability. Our goal was to provide a pragmatic adjunct to transvaginal ultrasonography and CA-125 in equivocal cases, supporting risk-adapted, individualized decision-making.

## 2. Materials and Methods

### 2.1. Study Design and Population

We conducted a retrospective cohort study at the Department of Obstetrics and Gynecology, University of Debrecen (Hungary), including postmenopausal (POM) patients who underwent surgery for an adnexal cystic lesion and had complete preoperative ultrasound and laboratory data available. Postmenopause was defined as ≥12 months of amenorrhea or prior hysterectomy performed after the age of 50. Exclusion criteria were as follows: non-cystic adnexal masses, pregnancy, active infection or systemic inflammatory disease at blood draw, concurrent malignancy other than ovarian, and missing key variables (ultrasound morphology or blood counts). The study was approved by the institutional review board (DE RKEB 4606-2016, approved: 14 January 2017), with a waiver of informed consent due to the retrospective design.

### 2.2. Ultrasound Acquisition and Morphology Definition

Preoperative transvaginal ultrasound (TVUS) examinations were performed by experienced operators using high-frequency endovaginal transducers: ATL HDI-3000 system (Bothell, WA, USA; 5.9 MHz probe) or a Medison Accuvix XQ (Medison Co., Seoul, Republic of Korea; 5–8 MHz probe; Kretztechnik AG, Zipf, Austria). Imaging followed department protocols.

Based on ultrasound morphology, cysts were classified into two groups (representative examples are provided in Figure 1):

1.Simple cysts—Unilocular, anechoic cysts with smooth walls and no papillary projections.2.Complex cysts—Cystic lesions with additional features such as septations, solid components, internal echogenicity, or vascularity.

### 2.3. Biomarkers and Calculations

Ultrasound findings—including morphology and lesion size—were correlated with final histopathological diagnoses and preoperative serum CA-125 levels, neutrophil-to-lymphocyte ratio (NLR), and platelet-to-lymphocyte ratio (PLR). The cut-off value for CA-125 was set at 35 kIU/L. The postmenopausal cohort was further subdivided by cyst size using a 5 cm threshold.

Preoperative CA-125 levels were measured using a chemiluminescent microparticle immunoassay (Abbott ARCHITECT, Abbott Laboratories, Abbott Park, IL, USA). Neutrophil-to-lymphocyte ratio (NLR) and platelet-to-lymphocyte ratio (PLR) were calculated using the following formulas: NLR = absolute neutrophil count/absolute lymphocyte count; PLR = platelet count/absolute lymphocyte count. These values were derived from complete blood counts performed on automated hematology analyzers (Sysmex XN-Series, Kobe, Japan).

### 2.4. Reference Standard and Outcomes

The primary outcome was malignancy on final histopathology (all epithelial types), adjudicated according to contemporary criteria. Benign pathology included functional and non-neoplastic cysts and benign neoplasms.

### 2.5. Statistical Analysis

All analyses were restricted to the POM cohort. Continuous variables were inspected for distribution and summarized as mean ± SD or median (IQR); between-group comparisons used Student’s *t*-test or Mann–Whitney U test, as appropriate. Categorical variables were compared with χ^2^ or Fisher’s exact test. Prespecified subgroup descriptions were performed within POM by (i) morphology (simple vs. complex) and (ii) size (<5 cm vs. ≥5 cm). To evaluate independent associations with malignancy, we fitted a multivariable logistic regression model including age, cyst size, morphology, CA-125, NLR, and PLR. Linearity in the logit for continuous predictors was assessed and transformations or categorization applied if violated. Multicollinearity was checked (variance inflation factors). Model results are reported as adjusted odds ratios (aOR) with 95% confidence intervals and *p*-values. For descriptive diagnostic metrics (specificity and NPV), we applied literature-based thresholds (NLR ≈ 2.36; PLR ≈ 150) as reported by Yun et al. [24]. Two-sided *p* < 0.05 was considered statistically significant. Analyses were performed in SPSS v10.0 (IBM Corp., Armonk, NY, USA).

## 3. Results

Preoperative transvaginal ultrasound data were obtained in 103 patients (mean age: 61.8 years; range: 45–88 years). According to morphological diagnosis, 27 cysts were simple and 76 complex. Among simple cysts, 6 measured <5 cm and 21 measured ≥5 cm in their largest dimension. Ten and sixty-six of the complex cysts were <5 cm and ≥5 cm, respectively. Of these, 29 (28.2%) were histologically malignant: 28 in the complex cyst group and 1 in the simple cyst group. The only malignant simple cyst was greater than 5 cm in size. This difference in percentage of malignancy in the different morphologic types was statistically significant (Fisher exact test, *p* = 0.001). Figure 2 shows the distribution of cyst morphology by histology.

Figure 3 shows the comparison of systemic inflammatory markers (NLR, PLR) between malignant and benign lesions. Results showed that both NLR and PLR were significantly higher in the malignant group when compared to the benign group (*p* < 0.001 and *p* < 0.001, respectively). Patient age was also significantly higher in the malignant group (*p* = 0.019), and cyst size was not significantly different (*p* = 0.248).


Analysis of the diagnostic performance parameters demonstrated that NLR and PLR showed a high negative predictive value (NPV) and specificity, especially in cysts with simple morphology. NPVs were between 0.94 and 1.00 with specificities of 76.5–90.9%. These descriptive diagnostic metrics were calculated using literature-based thresholds (NLR ≈ 2.36; PLR ≈ 150) from Yun et al. [24].

Table 1 shows the distribution of ultrasound morphology in benign and malignant lesions among patients. The median and IQR values of the NLR and PLR, patient age, and cyst size parameters are demonstrated in Table 2 for all, benign, and malignant cases. It shows the power of NLR and PLR in distinguishing benign from malignant masses regardless of patient age and the size of the adnexal mass. Odds ratios for the association between malignancy and the various parameters are demonstrated in Table 3, with complex morphology having the most significant association (OR 11.93; *p* = 0.02).

Those findings highlight the importance of grayscale ultrasound morphology and inflammatory factors in ovarian cyst risk stratification.

In addition to the observed differences in NLR and PLR values, we performed subgroup analyses to assess whether these markers retained diagnostic relevance across different cyst morphologies and sizes. Among complex cysts, elevated NLR and PLR values were significantly associated with malignancy (*p* < 0.001), suggesting that these systemic inflammatory markers may reinforce ultrasound findings in equivocal cases. Conversely, among simple cysts, NLR and PLR values were generally lower and did not demonstrate significant discriminatory power.

Furthermore, we stratified the cohort by lesion size (<5 cm vs. ≥5 cm). In both subgroups, median NLR and PLR values remained higher in malignant lesions, although the statistical power was greater in the ≥5 cm group due to higher event rates. Notably, when stratifying by both morphology and size, the subgroup of complex, large (≥5 cm) cysts showed the highest proportion of malignancy and the highest inflammatory marker values. This supports the hypothesis that complex morphology combined with increased size and elevated inflammatory markers may warrant more aggressive diagnostic follow-up.

Stratified summaries by morphology and size are presented in Table 3. Across strata, malignant lesions showed higher NLR and PLR than benign lesions; differences were most pronounced in complex cysts—particularly those ≥5 cm—while distinctions were minimal in simple cysts where malignancy was uncommon. Table 3 provides the first stratum-specific summary of NLR and PLR by morphology and size in a postmenopausal cohort. In simple cyst strata, malignancy was rare, precluding valid within-stratum hypothesis testing; therefore, *p*-values are NA where group sizes did not permit exact comparisons.

In the multivariable model, complex morphology remained independently associated with malignancy (OR 11.93; *p* = 0.02), whereas NLR showed a positive but non-significant association (OR 1.43; *p* = 0.14); PLR was not independently associated (*p* = 0.66).

This further underscores the additive value of NLR in preoperative assessment. PLR, while significant in univariate analysis, did not maintain independent predictive power in the multivariable model, potentially reflecting interindividual variability or confounding factors (Table 4).

These extended findings highlight the nuanced relationship between lesion characteristics, systemic inflammation, and malignancy risk, reinforcing the utility of a multimodal diagnostic approach.

## 4. Discussion

### 4.1. Principal Findings

In this postmenopausal cohort, grayscale morphology remained the cornerstone of triage: complex cysts were strongly associated with malignancy, whereas simple cysts rarely harbored cancer. Systemic inflammatory indices derived from routine blood counts were higher in malignant than benign lesions. In multivariable analysis, complex morphology remained independently associated with malignancy, while NLR showed a positive but non-significant trend and PLR was not independently associated.

### 4.2. Clinical Implications

To the best of our knowledge, this is among the first studies to show that values of the neutrophil-to-lymphocyte ratio (NLR) and the platelet to lymphocyte ratio (PLR) can be used as reliable adjuncts to ultrasound for the assessment of ovarian cystic lesions in postmenopausal women. Our observations strongly indicate that these systemic inflammatory markers are promising adjuncts for distinguishing benign from malignant cystic lesions and could become complementary diagnostic tools alongside conventional imaging techniques.

### 4.3. Comparison with Existing Literature

According to the GLOBOCAN 2020 database, ovarian cancer (OC) accounted for 313,959 new cases and 207,252 deaths worldwide. It remains the second most common gynecologic malignancy and the fifth leading cause of cancer-related mortality among women [38,39]. Ovarian cysts in postmenopausal women make a unique diagnostic challenge due to their increased likelihood of malignancy compared to premenopausal cysts [4]. The differentiation between benign and malignant lesions is crucial, as early-stage ovarian cancer is often asymptomatic and difficult to detect. Current clinical guidelines emphasize the need for reliable diagnostic tools to stratify risk and ensure appropriate patient management, especially in those areas where expert ultrasonographers and high-end ultrasound equipment are not available.

The evaluation of ovarian cysts traditionally relies on transvaginal ultrasound (TVUS), CA-125 levels, and in some cases MRI [14]. Scoring systems such as the International Ovarian Tumour Analysis (IOTA) Simple Rules and ADNEX model have been developed to improve accuracy [40]. However, while these methods provide valuable information, they have limitations in sensitivity and specificity, especially in differentiating borderline cases [41]. Age and cyst morphology are key factors in malignancy risk stratification. Complex cystic structures, multilocular solid tumors, and the presence of papillary projections are associated with a higher likelihood of malignancy [14]. Our study supports previous findings that malignancy rates increase with age and, as a result of this, postmenopausal women exhibit a higher prevalence of complex cysts and a greater risk of ovarian cancer as well [4].

Our results demonstrate significantly higher NLR and PLR values in malignant ovarian cysts compared to benign lesions. These findings align with previous studies suggesting that elevated systemic inflammatory markers correlate with tumor progression and poor prognosis [42]. High NLR and PLR levels reflect the tumor microenvironment’s inflammatory state, which contributes to tumor angiogenesis and immune evasion. The ability to rule out malignancy with confidence is critical in clinical practice to avoid unnecessary surgical interventions. We observed that lower NLR and PLR values were strongly associated with benign ovarian cysts, indicating their potential role in excluding malignancy [28]. This finding is particularly valuable in reducing patient anxiety and optimizing surgical decision-making [43]. Despite the fact that Huang et al. [31] recently demonstrated that combining CA-125 with NLR and PLR substantially improves the diagnostic performance in differentiating benign and borderline/malignant epithelial tumors, this connection has not been examined in our study.

A high NPV ensures that patients with low-risk profiles are accurately identified, reducing unnecessary surgeries and associated morbidity. Our study highlights that both NLR and PLR exhibited strong negative predictive values, suggesting their utility as non-invasive biomarkers for triaging patients [23]. Being accessible and cost-effective, these markers provide a practical add-on to existing imaging-based risk assessment strategies.

Our findings suggest that integrating NLR and PLR into routine diagnostic workflows could enhance the accuracy of ovarian cyst evaluation. These markers can be particularly useful when combined with imaging findings, aiding clinicians in refining risk stratification and improving patient management [43]. Combining transvaginal ultrasound with multi-biomarker panels has demonstrated improved differentiation between benign and malignant ovarian masses, suggesting a more effective approach for preoperative evaluation [44]. Future studies should explore the development of standardized cut-off values to facilitate their implementation in clinical practice [45]. Importantly, our results suggest that the integration of these markers with grayscale morphology enhances the robustness of diagnostic stratification. In a clinical scenario, a postmenopausal patient with a complex cyst and elevated NLR/PLR values may warrant expedited surgical referral, whereas a similar morphological finding with normal inflammation markers may prompt short-term surveillance and multidisciplinary review. This tiered approach supports a broader clinical trend toward risk-adapted, personalized medicine.

NLR and PLR likely capture elements of the tumor–host inflammatory response. The stronger multivariable signal for NLR compared with PLR in our cohort may reflect differences in biologic responsiveness or greater collinearity of PLR with other variables. We did not attempt to derive new cut-offs; thresholds reported across studies vary widely. Instead, we emphasize the direction and independence of associations and the practical integration of these indices with morphology rather than reliance on any single value.

### 4.4. Strengths and Limitations

Strengths include histology as the reference standard, a clearly defined postmenopausal target population with higher pretest probability, standardized morphology definitions, and evaluation of independent effects in multivariable models.

Limitations include the retrospective, single-center design; a modest number of malignant cases; potential confounding of inflammatory indices by intercurrent conditions (e.g., occult infection, autoimmune disease, cardiovascular comorbidity); single preoperative laboratory measurements; and lack of formal inter-observer assessment for ultrasound.

We intentionally restricted analyses to postmenopausal women to preserve internal validity; therefore, results should not be generalized to premenopausal patients without prospective evaluation. We also did not perform discrimination analyses aimed at threshold optimization; our objective was to test independent associations and provide a pragmatic adjunct to established ultrasound pathways.

### 4.5. Future Directions

We envision several directions for future work. Prospective studies with larger, more diverse populations are needed to validate our findings and establish optimal thresholds. Integration with IOTA models or artificial intelligence-based image analysis may offer further refinement of diagnostic algorithms. In particular, modeling efforts could quantify the incremental value of NLR and PLR when added to existing clinical predictors. Finally, given the encouraging NPVs observed, future research should examine whether inflammatory markers can support conservative management strategies in select postmenopausal women with low-risk features.

## 5. Conclusions

In postmenopausal women with adnexal cysts, grayscale morphology remains the anchor of diagnostic triage. Systemic inflammatory indices—especially NLR—provide an accessible, complementary signal that strengthens risk assessment when features are equivocal. Integrating these markers into ultrasound-centered workflows may support more consistent, individualized decision-making without adding cost or complexity.

## Figures and Tables

**Figure 1 diagnostics-15-02220-f001:**
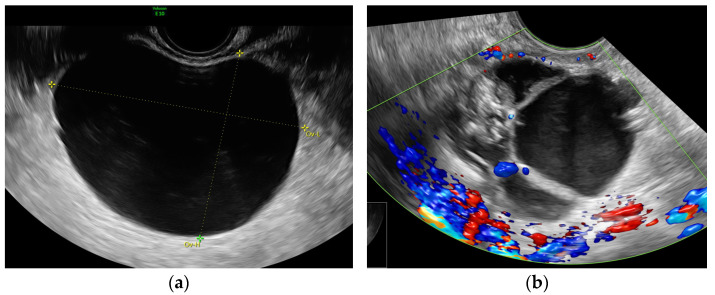
Representative images of ovarian cyst morphology. (**a**) **Simple cyst:** Unilocular, anechoic, smooth walled regular cyst. (**b**) **Complex cyst:** Multilocular-solid morphology, vessels are detected in the cyst wall, low-level echogenicity is present in the cyst content.

**Figure 2 diagnostics-15-02220-f002:**
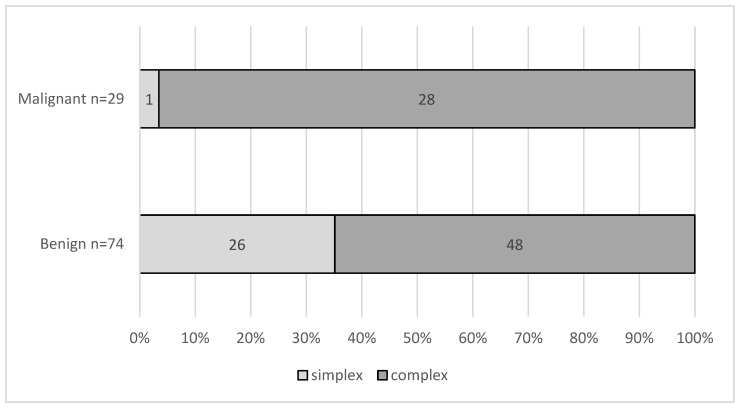
Distribution of cyst morphology (simple vs. complex) by histology (benign vs. malignant) in the postmenopausal cohort (*n* = 103).

**Figure 3 diagnostics-15-02220-f003:**
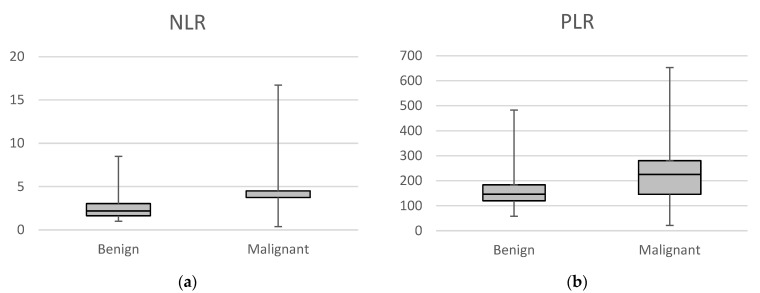
Box plots comparing NLR and PLR between benign and malignant cases (**a**) **NLR** values; (**b**) **PLR** values.

**Table 1 diagnostics-15-02220-t001:** Distribution of patient data in benign and malignant groups.

Outcome (Fisher’s Exact Test, *p* = 0.001)			
	Benign	Malignant	Total
**Simple N**	26	1	27
**Simple %**	96.3	3.7	100
**Total %**	35.14	3.45	26.21
**Complex N**	48	28	76
**Complex %**	63.16	36.84	100
**Total %**	64.86	96.55	73.79
**Total N**	74	29	103
**Total %**	71.84	28.16	100

**Table 2 diagnostics-15-02220-t002:** Median and IQR values of NLR and PLR. Age and size for all patients and benign and malignant masses.

	All Patient Data	Benign	Malignant	Benign vs. Malignant *p*-Value
NLR	2.31 (1.78–3.72)	2.18 (1.62–3.04)	3.74 (2.63–4.5)	<0.001
PLR	150.57 (123.13–218.44)	145.9 (119.57–184.23)	225.33 (145.54–280.43)	<0.001
Age	62 (55–67)	60.5 (55–66)	65 (59–72)	0.019
Size	70 (53–120)	70 (55–106)	85 (53–130)	0.248

**Table 3 diagnostics-15-02220-t003:** NLR and PLR by morphology and size strata in the postmenopausal cohort. *p*-values: Mann–Whitney U for NLR/PLR within stratum; NA where one group had zero events.

	NLR Median (IQR)	PLR Median (IQR)	*n*	Benign *n* (%)	Malignant *n* (%)	*p*-Value (NLR) Benign vs. Malignant	*p*-Value (PLR) Benign vs. Malignant
Simple < 5 cm	2.16 (2.05–2.44)	130.44 (108.59–172.46)	6	6 (100)	0 (0)	<0.001	<0.001
Simple ≥ 5 cm	2.04 (1.61–2.62)	130.77 (111.26–175.14)	21	20 (95.24)	1 (4.76)		
Complex < 5 cm	2.48 (2.17–3.74)	172.31 (120.10–248.48)	10	8 (80)	2 (20)		
Complex ≥ 5 cm	2.68 (1.75–3.82)	158.04 (130.74–225.33)	66	40 (60.61)	26 (39.39)		

**Table 4 diagnostics-15-02220-t004:** Multivariable logistic regression for malignancy (postmenopausal cohort, *n* = 103).

	OR	*p* Value
NLR	1.43	0.14
PLR	1.00	0.66
Age	1.05	0.09
Size	1.00	0.95
Complex/Simple	11.93	0.02

## Data Availability

The raw data supporting the conclusions of this article will be made available by the authors on request.

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
