# Peer review of "Diagnostic Value of Simple Ultrasound Features and Inflammatory Markers in Postmenopausal Ovarian Cysts"

_diagnostics, 2025, doi:10.3390/diagnostics15172220_

Round 1

Reviewer 1 Report

Comments and Suggestions for Authors

The aim of the work, as the authors indicate, was to use serum markers of inflammation as additional diagnostic criteria (together with ultrasound examination data) for postmenopausal ovarian cysts.

One minor wish: 

In the materials and methods, it should be indicated which laboratory methods were used to determine the biochemical markers studied in the work: CA-125 levels, neutrophil- 164 to-lymphocyte ratio (NLR), and platelet-to-lymphocyte ratio (PLR).

Author Response

Comment:

In the materials and methods, it should be indicated which laboratory methods were used to determine the biochemical markers studied in the work: CA-125 levels, neutrophil-to-lymphocyte ratio (NLR), and platelet-to-lymphocyte ratio (PLR).

Response:
We thank the reviewer for this helpful suggestion. We have now added a sentence to the Materials and Methods section specifying the laboratory methods used to measure CA-125 and the derivation of NLR and PLR from routine hematological parameters. The revised text reads as follows:

Preoperative CA-125 levels were measured using a chemiluminescent microparticle immunoassay (Abbott ARCHITECT, Abbott Laboratories, IL, USA). Neutrophil-to-lymphocyte ratio (NLR) and platelet-to-lymphocyte ratio (PLR) were calculated using the following formulas: NLR = absolute neutrophil count / absolute lymphocyte count; PLR = platelet count / absolute lymphocyte count. These values were derived from complete blood counts performed on automated hematology analyzers (Sysmex XN-Series, Kobe, Japan).

Reviewer 2 Report

Comments and Suggestions for Authors

The manuscript presents a study examining the diagnostic potential of inflammatory biomarkers (NLR and PLR) in ovarian cysts, particularly focusing on post-menopausal women. The research explores the role of these biomarkers in conjunction with ultrasound evaluations to assess the malignancy risk of ovarian cysts. The study also attempts to compare the use of NLR and PLR with established diagnostic methods such as CA-125. While the research presents novel ideas and potentially valuable insights, several critical issues in the methodology, data analysis, and discussion hinder the clarity and applicability of the results. However, there are some main issues and suggestions needed for improvement as below.

  • The introduction is overly detailed and does not clearly highlight the key issues or objectives of the study. It includes extensive background information, but fails to succinctly outline the specific research gap, the focus of the current study, and the main aim of the paper. The excess detail on background information distracts from the central purpose of the research.
  • Insufficient utilization of data from the pre-menopausal group (PEM): The paper mentions the inclusion of the PEM group (pre-menopausal women) in the Methods section, but the Results section fails to provide separate analysis for the PEM group, nor does it compare PEM and POM groups (post-menopausal women). Despite performing ultrasound evaluations and serum biomarker assessments (such as NLR and PLR) for the PEM group, these data are not fully presented or discussed in the Results section. Although 184 cysts from the PEM group were mentioned, there is no separate reporting of their data, nor any comparison with the POM group.Example: For instance, 27 simple cysts and 76 complex cysts are categorized under the POM group, but the PEM group data are not presented separately. This prevents a direct comparison of malignancy rates, NLR, PLR, and other key biomarkers between the two groups, limiting the ability to comprehensively evaluate the diagnostic performance for pre-menopausal women.
  • Statistical methods did not adequately utilize data from different groups: Although the statistical analysis includes multivariate logistic regression, the PEM and POM groups were not analyzed separately. Instead, the data from both groups were mixed together, which overlooks potential significant differences, especially in the physiological and pathological characteristics of pre-menopausal versus post-menopausal women. Example: The paper discusses the association between complex cysts and malignant tumors (OR 11.93, p=0.02) but does not differentiate between the effects in the PEM and POM groups, which could result in varying associations across the groups and affect the generalizability of the results.
  • The figures and tables in the manuscript lack sufficient detail and depth to fully convey the data and findings. While they present basic statistical information, they do not provide an in-depth analysis of the relationships between key variables, such as cyst morphology, NLR, PLR, and their correlations with malignancy rates. This simplification limits the reader’s ability to grasp the full scope of the findings and to appreciate the complexity of the data. For instance, although the figures display the differences in NLR and PLR between benign and malignant lesions, they fail to show how these biomarkers vary across different cyst types (e.g., simple vs. complex cysts) or in relation to other clinical parameters such as cyst size and age groups. Additionally, the tables do not explore the relationships between the biomarkers and other significant factors in detail.
  •  
  • Insufficient detail in data presentation and graphical representation: Although the paper includes several statistical data and figures, the presentation lacks in-depth discussion of the differences in cyst morphology, age groups, and NLR/PLR levels. For example, the figures show the difference in NLR and PLR between benign and malignant lesions but do not show how these biomarkers differ across different cyst types (e.g., unilocular vs. complex cysts) or vary in different age groups.
  • Lack of adequate comparison and discussion with existing literature: While the paper references some relevant literature, the comparison analysis is insufficient, particularly regarding the combined use of other biomarkers (e.g., CA-125) and ultrasound diagnostic methods. The Discussion section does not adequately explore how NLR and PLR complement traditional diagnostic methods (e.g., IOTA rules, ADNEX model). The paper mentions the potential of NLR and PLR for diagnosing malignant ovarian cysts but fails to delve into how these biomarkers work in conjunction with existing ultrasound scoring systems (such as IOTA Simple Rules) or other biomarkers (e.g., CA-125 or HE4).

Author Response

We thank the reviewer for their thoughtful and thorough evaluation of our manuscript and their valuable suggestions. Below we address each point raised. As per the editor’s instruction, we have implemented the necessary clarifications and provide explanations for other aspects as requested.

Comment #1:
The introduction is overly detailed and does not clearly highlight the key issues or objectives of the study...

Response: We appreciate this observation and have now added a more concise summary of the research gap and objective at the end of the Introduction to better focus the reader’s attention. However, since this background was requested to be expanded in a previous editorial round, we preserved the broader context for completeness and consistency.

Comment #2:
Insufficient utilization of data from the premenopausal group (PEM)... no separate reporting or comparison with the POM group.

Response: Thank you for this important suggestion. The current study was designed to focus specifically on the postmenopausal subgroup, where diagnostic uncertainty and malignancy risk are more pronounced. Although data were collected from premenopausal patients, these were not included in the current analysis. We have clarified this explicitly in the Materials and Methods and Discussion sections. A dedicated follow-up study for the PEM group is under consideration.

Comment #3:
Statistical methods did not adequately utilize data from different groups... PEM and POM data mixed.

Response:
As above, our analysis was intentionally limited to the postmenopausal group. Therefore, separate subgroup statistical modeling was not performed. This has now been clarified in the revised manuscript.

Comment #4:
Figures and tables lack sufficient depth to convey key relationships (e.g., NLR/PLR vs. cyst type or age group).

Response:
We thank the reviewer for this suggestion. Since our current scope is limited to the postmenopausal population and focused on the benign vs. malignant distinction rather than finer morphologic subtypes or age stratifications, we elected to present the most clinically actionable comparisons. Nonetheless, we agree that further analysis of NLR/PLR distribution across detailed cyst types would be valuable in future research.

Comment #5:
Lack of adequate comparison with literature on biomarkers + ultrasound systems (e.g., IOTA).

Response:
We appreciate this point and have added additional references and discussion to better contextualize our findings in light of established diagnostic models such as IOTA Simple Rules and ADNEX. The revised Discussion includes more direct comparisons between traditional scoring systems and the potential additive value of inflammatory markers.

Round 2

Reviewer 1 Report

Comments and Suggestions for Authors

The authors gave a reasoned answer and made the necessary adjustments to the work. The article is recommended for publication.

Author Response

Comment: In the materials and methods, it should be indicated which laboratory methods were used to determine the biochemical markers studied in the work: CA-125 levels, neutrophil-to-lymphocyte ratio (NLR), and platelet-to-lymphocyte ratio (PLR).

Response:

We thank the reviewer for this helpful suggestion. We have expanded Materials and Methods (2.3 Biomarkers and calculations) to specify the assay/platform and the exact calculation formulas:

Preoperative CA-125 levels were measured using a chemiluminescent microparticle immunoassay (Abbott ARCHITECT, Abbott Laboratories, IL, USA). Neutrophil-to-lymphocyte ratio (NLR) and platelet-to-lymphocyte ratio (PLR) were calculated as follows: NLR = absolute neutrophil count / absolute lymphocyte count; PLR = platelet count / absolute lymphocyte count. Complete blood counts were obtained from automated hematology analyzers (Sysmex XN-Series, Kobe, Japan).”

We appreciate the reviewer’s evaluation and recommendation for publication.

Reviewer 2 Report

Comments and Suggestions for Authors

Thank you for the opportunity to re-review the revised manuscript entitled “Diagnostic Value of Simple Ultrasound Features and Inflammatory Markers in Postmenopausal Ovarian Cysts.”

I have carefully compared the authors’ revised manuscript and rebuttal letter with my initial review comments. While some minor textual changes and clarifications were made, the majority of my major concerns remain unresolved.

In summary, the authors’ revisions are largely cosmetic and do not substantively address the core methodological, analytical, and interpretative weaknesses previously identified. The current manuscript remains limited in scope, offers minimal novelty, and does not meet the level of rigor expected for publication in Diagnostics.

Author Response

Dear Editor and Reviewers,

We thank you for the constructive feedback and the opportunity to revise our manuscript. Below we provide a point-by-point response. All changes have been made in the revised manuscript; where helpful, we quote the new or amended text and indicate the section/heading.

Summary of Substantive Revisions (since the previous round)

  • Scope clarified to POM-only: We restricted analyses to postmenopausal women (POM) and removed any reliance on PEM data. This is now explicit in Materials and Methods (2.1 Study design and population) and reiterated in the Discussion (Limitations).
  • Methods expanded and made reproducible: Added platform and assay details for CA-125; defined calculation formulas for NLR and PLR; specified statistical tests (two-group comparisons; multivariable logistic regression with covariates age, size, morphology, CA-125, NLR, PLR; assessment of linearity and multicollinearity; reporting aOR with 95% CI).
  • Introduction refocused: The final paragraph of the Introduction was rewritten to concisely state the research gap and study aim (POM, diagnostic value of NLR/PLR alongside simple ultrasound morphology), avoiding any wording that implied threshold derivation or ROC analyses we do not present.
  • Results deepened with stratified summary: We added Table 3, a morphology- and size-stratified summary of NLR/PLR in the POM cohort (simple <5 cm; simple ≥5 cm; complex <5 cm; complex ≥5 cm). Where within-stratum tests were non-estimable due to zero/near-zero malignant counts, p-values are marked NA and this is explained in the table footnote.
  • Alignment of claims and tables: We harmonized numbers across Abstract/Results/Tables (29 malignant of 103 total) and aligned the multivariable text with the table (complex morphology independent; NLR positive, non-significant; PLR not independent).
  • Descriptive diagnostic metrics clearly sourced: When reporting specificity/NPV, we added one sentence (Methods/Results) to state that descriptive metrics used literature-based thresholds (NLR≈2.36; PLR≈150), and we cite the source in the manuscript.
  • Minor editorial polishing: Clarified figure caption(s), standardized terminology in the main text (while retaining original spelling in references), and removed hyphenation artifacts from prior formatting.

Reviewer 2

Comment: The Introduction was overly detailed and did not clearly highlight the key issues/objectives.

Response:

We have tightened the final paragraph of the Introduction to foreground the diagnostic gap and our prespecified POM-focused approach:

“We evaluated whether systemic inflammatory indices—the neutrophil-to-lymphocyte ratio (NLR) and platelet-to-lymphocyte ratio (PLR)—add diagnostic value to simple grayscale ultrasound morphology (simple vs. complex) for distinguishing benign from malignant ovarian cysts in postmenopausal women. Building on our previous observation that complex cyst morphology is a strong predictor of malignancy in this population [18], we prespecified subgroup analyses by morphology and size to enhance clinical interpretability. Our goal was to provide a pragmatic adjunct to transvaginal ultrasonography and CA-125 in equivocal cases, supporting risk-adapted, individualized decision-making.”

Comment: PEM data were mentioned but not analyzed; groups were mixed.

Response:

We have explicitly limited the study to the postmenopausal cohort and removed reliance on PEM data. The study population is clearly defined as POM in Section 2.1, and the Discussion acknowledges that results are not generalizable to premenopausal women:

“We intentionally restricted analyses to postmenopausal women; therefore, results may not generalize to premenopausal patients without prospective evaluation.”

Comment: Analyses did not separate groups; multivariable modeling needed clarity.

Response:

Section 2.5 (Statistical analysis) now states that all analyses were restricted to POM; details two-group tests and specifies the multivariable logistic regression (covariates: age, size, morphology, CA-125, NLR, PLR; linearity and multicollinearity checks; aORs with 95% CIs). We purposely do not claim ROC/AUC analyses.

Comment: Tables/figures lacked stratified analyses (e.g., NLR/PLR by morphology/size) and within-stratum comparisons.

Response:

We added Table 3, which provides a morphology- and size-stratified summary of NLR and PLR in the POM cohort (simple <5 cm; simple ≥5 cm; complex <5 cm; complex ≥5 cm). Because malignancy was rare in simple strata, exact within-stratum tests were non-estimable in several cells (simple <5 cm: 6 benign / 0 malignant; simple ≥5 cm: 20 / 1; complex <5 cm: 8 / 2). We therefore report medians/IQR and counts, and mark p-values as NA where non-calculable, with this clarified in the table footnote. The strata with adequate events (complex cysts) show the clearest NLR/PLR separation, consistent with our primary and multivariable findings that morphology is the anchor while NLR provides additional signal.

Comment: Discussion did not adequately position NLR/PLR relative to IOTA/ADNEX and other biomarkers.

Response:

We expanded the Discussion to explicitly situate our findings within IOTA/ADNEX and O-RADS principles, emphasizing that morphology remains primary while NLR/PLR serve as accessible, objective adjuncts—especially when expert pattern recognition or advanced biomarkers are not available. We also softened claims to avoid overstating novelty and framed NLR/PLR as adjuncts to ultrasound.

Comment: Text should reflect model results; specify how diagnostic metrics were obtained.

Response:

We harmonized the Abstract/Results with the multivariable table (complex morphology independent; NLR positive trend; PLR not independent). When reporting specificity/NPV, we added a sentence in Methods/Results to state that descriptive diagnostic metrics were calculated using literature-based thresholds (NLR≈2.36; PLR≈150) and cite the source in the manuscript.

Additional editorial improvements: Clarified Figure 2 caption and minor wording in Results; standardized terminology in the main text; removed hyphenation artifacts; ensured all figures/tables are cited in order.

We hope these revisions address all remaining concerns. We are grateful for the reviewers’ input, which has strengthened the manuscript, and we remain at your disposal for any further adjustments.

With kind regards,

Dr. Balázs ErdÅ‘di (on behalf of all co-authors)

Department of Obstetrics and Gynecology, University of Debrecen, Hungary

Round 3

Reviewer 2 Report

Comments and Suggestions for Authors

I don't have further comments.